# Fully Online Meta-Learning Without Task Boundaries

## Abstract

While deep networks can learn complex functions such as classifiers, detectors, and trackers, many applications require models that continually adapt to changing input distributions, changing tasks, and changing environmental conditions. Indeed, this ability to continuously accrue knowledge and use past experience to learn new tasks quickly in continual settings is one of the key properties of an intelligent system. For complex and high-dimensional problems, simply updating the model continually with standard learning algorithms such as gradient descent may result in slow adaptation. Meta-learning can provide a powerful tool to accelerate adaptation yet is conventionally studied in batch settings. In this paper, we study how meta-learning can be applied to tackle online problems of this nature, simultaneously adapting to changing tasks and input distributions and meta-training the model in order to adapt more quickly in the future. Extending meta-learning into the online setting presents its own challenges, and although several prior methods have studied related problems, they generally require a discrete notion of tasks, with known ground-truth task boundaries. Such methods typically adapt to each task in sequence, resetting the model between tasks, rather than adapting continuously across tasks. In many real-world settings, such discrete boundaries are unavailable, and may not even exist. To address these settings, we propose a Fully Online Meta-Learning (FOML) algorithm, which does not require any ground truth knowledge about the task boundaries and stays fully online without resetting back to pre-trained weights. Our experiments show that FOML was able to learn new tasks faster than the state-of-the-art online learning methods on Rainbow-MNIST, and CIFAR100 datasets.

## 1 Introduction

Flexibility and rapid adaptation are a hallmark of intelligence: humans can not only solve complex problems, but they can also figure out *how* to solve them very rapidly, as compared to our current machine learning algorithms. Such rapid adaptation is crucial for both humans and computers: for humans, it is crucial for survival in changing natural environments, and it is also crucial for agents that classify photographs on the Internet, interpret text, control autonomous vehicles, and generally make accurate predictions with rapidly changing real-world data. While deep neural networks are remarkably effective for learning and representing *accurate* models (He et al., 2015; Krizhevsky et al., 2012; Simonyan & Zisserman, 2014; Szegedy et al., 2015), they are comparatively unimpressive when it comes to adaptability, due to their computational and data requirements. Meta-learning in principle mitigates this problem, by leveraging the generalization power of neural networks to accelerate adaptation to new tasks (Finn et al., 2019; Li et al., 2017; Nichol et al., 2018; Nichol & Schulman, 2018; Park & Oliva, 2019; Antoniou et al., 2018). However, standard meta-learning algorithms operate in batch mode, making them poorly suited for continuously evolving environments. More recently, online meta-learning methods have been proposed with the goal of enabling continual adaptation (Finn et al., 2019; Jerfel et al., 2018; Yao et al., 2020; Nagabandi et al., 2018; Li & Hospedales, 2020), where a constant stream of data from distinct tasks is used for *both* adaptation and meta-training. In this scheme, meta-training is used to accelerate how quickly the network can adapt to each new task it sees, and *simultaneously* use that data from each new task for meta-training. This further accelerates how quickly each subsequent task can be acquired. However, current online meta-learning methods fall short of the goal of creating an effective adaptation system for online data

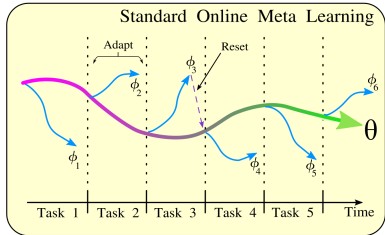 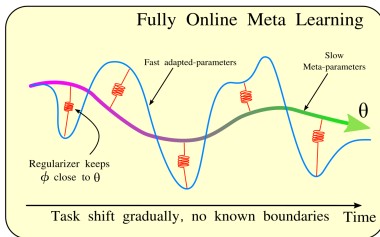

**Figure 1: Comparison of standard online meta-learning and FOML:** In standard online meta-learning (e.g., FTML (Finn et al., 2019)), shown on the left, adaptation is performed on one task a time, and the algorithm "resets" the adaptation process at task boundaries. For example, a MAML-based method would reset the current parameters back to the meta-trained parameters. In our approach (right), knowledge of task boundaries is not required, and the algorithm continually keeps track of *online* parameters $\phi$ and *meta*-parameters $\theta$. The online parameters are simply updated on the latest data, and the meta-parameters are updated to "pull" the online parameters toward fast-adapting solutions via a MAML-style meta-update.

in several ways: (1) they typically require task boundaries in the data stream to be known, making them ill-suited to settings where task boundaries are ill-defined and tasks change or evolve gradually, a common tread in real-world; (2) as a result, they typically re-adapt from the meta-trained model on each task, resulting in a very "discrete" mode of operation, where the model adapts to a task, then resets, then adapts to a new one. These limitations restrict the applicability of current online meta-learning methods to real-world settings. We argue that task boundary assumption is somewhat artificial in online settings, where the stream of incoming data is cleanly partitioned into discrete and well-separated tasks presented in sequence. In this paper, we instead develop a *fully* online meta-learning approach, which does not assume knowledge of task boundaries and does not re-adapt for every new task from the meta parameters.

Standard meta-learning methods consist of a meta-training phase, typically done with standard SGD, and an "inner loop" adaptation phase, which computes task specific parameter $\phi_i$ for the task $\mathcal{T}_i$ from a *support* set to make accurate predictions on a *query* set. For example, in model-agnostic meta-learning (MAML), adaptation consists of taking a few gradient steps on the support set, starting from the meta-trained parameter vector $\theta$, leading to a set of *post-adaptation* parameters, and meta-training optimizes the meta-trained parameters $\theta$ so that these gradient steps lead to good results. Previous extensions of such approaches into the online setting typically observe one task at a time, adapt to that task (i.e., compute post-adaptation parameters on it), and then *reset* $\phi_i$ back to the meta-trained parameters $\theta$ at the beginning of the next task. Thus, the algorithm repeatedly adapts, resets at the task boundary, adapts again, and repeats. This is illustrated in Figure 1 (left). However, in many realistic settings, the task boundaries are not known, and instead the tasks shift gradually over time. The discrete "resetting" procedure is a poor fit in such cases, and we would like to simply continue adapting the weights over time without ever resetting back to the meta-trained parameters, but still benefit from a concurrent meta-training process. For example, a meta-trained image-tagging model on the Internet (e.g., tagging friends in photographs) might gradually adapt to changing patterns and preferences of its users over time, where it would be unnatural to assume discrete shifts in what users want to tag. Similarly, a traffic prediction system might adapt to changing traffic patterns, including periodic changes due to seasons, and unexpected changes due to shifting economic conditions, weather, and traffic accidents. In this spirit, our method does not require any knowledge on the task boundaries as well as stays fully-online through out the learning.

The main contribution of our paper is FOML (fully online meta-learning), an online meta-learning algorithm that continually updates its online parameters with each new datapoint or batch of datapoints, while simultaneously performing meta-gradient updates on a separate set of meta-parameters using a buffer of previously seen data. FOML does not require ground truth knowledge of task boundaries, and does not reset the online parameters back to the meta-parameters between tasks, instead updating the online parameters continually in a fully online fashion. We compare FOML empirically to strong baselines and a state-of-the-art prior online meta-learning method, showing that FOML learns to adapt more quickly, and achieves lower error rates, both on a simple sequential image classification task from prior work and a more complex benchmark that we propose based on the CIFAR100 dataset, with a sequence of 1200 tasks.

## 2 RELATED WORK

Online meta-learning brings together ideas from online learning, meta learning, and continual learning, with the aim of adapting quickly to each new task while *simultaneously* learning how to adapt even more quickly in the future. We discuss these three sets of approaches next.

**Meta Learning:** Meta learning methods try to learn the high-level context of the data, to behave well on new tasks (*Learning to learn*). These methods involve learning a metric space (Koch et al., 2015; Vinyals et al., 2016; Snell et al., 2017; Yang et al., 2017), gradient based updates (Finn et al., 2017; Li et al., 2017; Park & Oliva, 2019; Nichol et al., 2018; Nichol & Schulman, 2018), or some specific architecture designs (Santoro et al., 2016; Munkhdalai & Yu, 2017; Ravi & Larochelle, 2016). In this work, we are mainly interested in gradient based meta learning methods for online learning. MAML (Finn et al., 2017) and its variants (Nichol et al., 2018; Nichol & Schulman, 2018; Li et al., 2017; Park & Oliva, 2019; Antoniou et al., 2018) first meta train the models in such a way that the meta parameters are close to the optimal task specific parameters (good initialization). This way, adaptation becomes faster when fine tuning from the meta parameters. However, directly adapting this approach into an online setting will require more relaxation on online learning assumptions, such as access to task boundaries and resetting back and froth from meta parameters. Our method does not require knowledge of task boundaries.

**Online Learning:** Online learning methods update their models based on the stream of data sequentially. There are various works on online learning using linear models (Cesa-Bianchi & Lugosi, 2006), non-linear models with kernels (Kivinen et al., 2004; Jin et al., 2010), and deep neural networks (Zhou et al., 2012). Online learning algorithms often simply update the model on the new data, and do not consider the past knowledge of the previously seen data to do this online update more efficiently. However, the online meta learning framework, allow us to keep track of previously seen data and with the "meta" knowledge we can update the online weights to the new data more faster and efficiently.

**Continual Learning:** A number of prior works on continual learning have addressed catastrophic forgetting (McCloskey & Cohen, 1989; Li & Hoiem, 2017; Ratcliff, 1990; Rajasegaran et al., 2019; 2020), removing the need to store all prior data during training. Our method does not address catastrophic forgetting for the meta-training phase, because we must still store all data so as to "replay" it for meta-training, though it may be possible to discard or sub-sample old data (which we leave to future work). However, our adaptation process is fully online. A number of works perform meta-learning for better continual learning, i.e. learning good continual learning strategies (Al-Shedivat et al., 2017; Nagabandi et al., 2018; Javed & White, 2019; Harrison et al., 2019; He et al., 2019; Beaulieu et al., 2020). However, these prior methods still perform batch-mode meta-training, while our method also performs the meta-training itself incrementally online, without task boundaries.

The closest work to ours is the follow the meta-leader (FTML) method (Finn et al., 2019) and its variants (Yao et al., 2020). FTML is a varaint of MAML that finetunes to each new task in turn, resetting to the meta-trained parameters between every task. While this method effectively accelerates acquisition of new tasks, it requires ground truth knowledge of task boundaries and, as we show in our experiments, our approach outperforms FTML *even when FTML has access to task boundaries and our method does not*. One of the limitations of these gradinet based meta learning methods is that, the memory requiment linearly increases with the number of inner loop updates. Ours has the same limitations as MAML and FTML, and have same order of memory requirement.. Further, there are some recent works which try to address Online meta learning Harrison et al. (2020); Denevi et al. (2019) in a similar setting to ours. Oonline-within-Online Denevi et al. (2019) tries to learn fast adaptation by learning from old tasks. However, OWO Denevi et al. (2019) requires the knowledge about the task boundaries, while our method does not need any extra information about the task boundaries. MOCA Harrison et al. (2020) address a very similar problem as ours, yet it tries to find the task boundaries or change points from past data. On the other hand our method can work even without estimating the change points. Additionaly Gupta et al. (2020); Caccia et al. (2020) address on solving continual learning with meta learning. Both works tries to minimize catastrophic forgetting, by precondition the meta gradients. While FOML focuses on rapidly learning new tasks, it can learn from 1000+ tasks, while continual learning methods usually has 10 to 100 tasks.

## 3 FOUNDATIONS

Prior to diving into online meta learning, we first briefly summarize meta learning, model agnostic meta-learning, and online learning in this section.

**Meta-learning:** Meta-learning address the problem of learning to learn. It uses the knowledge learned from previous tasks to quickly learn new tasks. Meta-learning assumes that the tasks are drawn from a stationary distribution $\mathcal{T} \sim \mathbb{P}(\mathcal{T})$. During the meta-training phase (outer-loop), $N$ tasks are assumed to be drawn from this distribution to produce the meta-training set, and the model is trained in such a way that, when a new task with its own training and test data $\mathcal{T} = \{\mathcal{D}_{\mathcal{T}}^{tr}, \mathcal{D}_{\mathcal{T}}^{te}\}$ is presented to it at meta-test time, the model should be able to adapt to this task quickly (inner-loop). Using $\theta$ to denote the meta-trained parameters, the meta-learning objective is:

$$\theta^* = \operatorname*{argmin}_{\theta} \mathbb{E}_{\mathcal{D}_{\mathcal{T}}^{tr} \text{ where } \mathcal{T} \sim \mathbb{P}(\mathcal{T})} \left[ \mathcal{L}(F_\theta(\mathcal{D}_{\mathcal{T}}^{tr}), \mathcal{D}_{\mathcal{T}}^{te}) \right], \tag{1}$$

where $F_\theta$ is the meta-learned adaptation process that reads in the training set $\mathcal{D}_t^{tr}$ and outputs task-specific parameters, prototypes, or features (depending on the method) for the new task $\mathcal{T}_i$.

**Model-agnostic meta-learning:** In MAML (Finn et al., 2017), the inner-loop function is (stochastic) gradient decent. Hence, during the MAML inner-loop adaptation, $F_\theta(\mathcal{D}_i^{tr})$ becomes $\theta - \alpha \nabla \mathcal{L}_\theta(\theta, \mathcal{D}_i^{tr})$ (or, more generally, multiple gradient steps). Intuitively, what this means is that meta-training with MAML produces a parameter vector $\theta$ that can quickly adapt to any task from the meta-training distribution via gradient descent on the task loss. One of the principle benefits of this is that, when faced with a new task that differs from those seen during meta-training, the algorithm "at worst" adapts with regular gradient descent, and at best is massively accelerated by the meta-training.

**Online learning:** In online learning, the model faces a sequence of loss functions $\{\mathcal{L}_t\}_{t=1}^{\infty}$ and a sequence of data $\{\mathcal{D}_t = \{(x, y)\}\}_{t=1}^{\infty}$ for every time step $t$. The function $f : x \to \hat{y}$ maps inputs $x$ to predictions $\hat{y}$. The goal of an online learning algorithm is to find a set of parameters for each time step $\{\phi\}_{t=1}^{\infty}$, such that the overall loss between the predictions $\hat{y}$ and the ground truth labels $y$ is minimized over the sequence. This is typically quantified in terms of regret:

$$\text{Regret}_T = \sum_{t=1}^{T} \mathcal{L}_t(\phi, \mathcal{D}_t) - \sum_{t=1}^{T} \mathcal{L}_t(\phi_t, \mathcal{D}_t). \tag{2}$$

where, $\phi_t = \operatorname{argmin}_\phi \mathcal{L}_t(\phi, \mathcal{D}_t)$. The first term measures the loss from the online model, and the second term measures the loss of the best possible model on that task. Various online algorithms try to minimize the regret as much as possible when introducing new tasks.

## 4 ONLINE META-LEARNING: PROBLEM STATEMENT AND METHODS

In an online meta-learning setting (Finn et al., 2019), the model $f_\phi$ observes datapoints one at a time from a online data stream $\mathcal{S}$. Each datapoint consists of an input $x_m^t$, where $t$ is the task index and $m$ is the index of the datapoint within that task, and a label $y_m^t$. The task changes over time and the model should be able to update the parameters $\phi$ to minimize the loss at each time step. *The goal of online meta-learning is to quickly learn each new task $\mathcal{T}_t$ and perform well as soon as possible according to the specified loss function.*

A simple baseline solution would be to just train the model on the current task $\mathcal{T}_t$. We denote this baseline as TFS (Train from Scratch). For every new task, the model simply trains a new set of parameters using all of the data from the current task $\mathcal{T}_t$ that has been seen so far:

$$\phi_{TFS}^t = \operatorname*{argmin}_{\phi} \frac{1}{M} \sum_{m=1}^{M} \mathcal{L}_t(\phi, (x_m^t, y_m^t)).$$

TFS has two issues. First, it requires the task boundaries to be known, which can make it difficult to apply to settings where this information is not available. Second, it does not utilize knowledge from other tasks, which greatly limits its performance even when task boundaries are available.

A straightforward way to utilize knowledge from other tasks of the online data stream is to store all the seen tasks in a large buffer $\mathcal{B}$, and simply keep training the model on all of the seen tasks. We will refer to this baseline method as TOE (Train on Everything):

$$\phi_{TOE}^t = \operatorname*{argmin}_{\phi} \frac{1}{Mt} \sum_{i=1}^{t} \sum_{m=1}^{M} \mathcal{L}_i(\phi, (x_m^i, y_m^i)).$$

TOE learns a function that fits all of the previously seen samples. However, this function may be far from optimal for the task at hand, because the different tasks may be mutually exclusive. Therefore, fitting a single model on all of the previously seen tasks might not provide a good task-specific model for the current task. A more sophisticated baseline, which we refer to as FTL (Follow the Leader), pre-trains a model on all of the previous tasks, and then fine-tunes it only on the data from the current task. Note that this is subtly different from FTL in the classic online learning setting, due to the difference in problem formulation. This can be achieved by initializing $\phi$ with pretrained weights up to the previous task $\phi_{TOE}^{t-1}$.

$$\phi_{FTL}^t = \underset{\phi}{\arg\min} \frac{1}{M} \sum_{m=1}^{M} \mathcal{L}_t(\phi_{TOE}^{t-1}, (x_m^t, y_m^t)).$$

Here, for the task $t$, we take a model that is pre-trained on all previously seen tasks ($f_{\phi_{TOE}^{t-1}}$) and fine-tune on the current task data. In this way, FTL can use the past knowledge to more quickly adapt to the new task. However, pre-training on past tasks may not necessarily result in an initialization that is conducive to fast adaptation (Finn et al., 2017; Nichol et al., 2018; Nichol & Schulman, 2018; Li et al., 2017). Finn *et al.* (Finn et al., 2019) proposed a MAML-based online meta-learning approach, where MAML is used to meta-train a "meta-leader" model on all previously seen tasks, which is then adapted on all data from the current task. This way, the meta-leader parameters will be much closer to new task optimal parameters, and because of this it is much faster to adapt to new tasks from the online data.

$$\phi_{FTML}^t = \underset{\phi}{\arg\min} \mathbb{E}_{(x_m^t, y_m^t) \sim T_t}[\mathcal{L}_t(\phi_{MAML}^{t-1}, (x_m^t, y_m^t))]$$

$$\text{where,} \ \ \phi_{MAML}^{t-1} = \underset{\phi}{\arg\min} \mathbb{E}_{T_j \sim \mathcal{D}(\mathcal{T})}[\mathcal{L}_j(\phi - \nabla \mathcal{L}_j(\phi, \mathcal{D}_j^{tr}), \mathcal{D}_j^{te})]$$

FTL and FTML try to efficiently use knowledge from past tasks to quickly adapt to the new task. However, pre-trained weights from FTL do not guarantee fast adaptation, and both methods require ground-truth knowledge of task boundaries. This knowledge may not be realistic in many real-world settings, where the tasks may change gradually and no external information is available to indicate task transitions. Although FTML can enable fast adaptation, the model needs to be "reset" at each task, essentially creating a "branch" on each task and maintaining two independent learning processes: an adaptation process, whose result is discarded completely at the end of the task, and a meta-training process, which does not influence the current task at all, and is only used for forward transfer into future tasks. For this reason, we argue that FTL and FTML are not fully online. In this work, our aim is to develop a *fully* online meta-learning method that continually performs both "fast" updates and "slow" meta-updates, does not need to periodically "reset" the adapted parameters back to the meta-parameters, and does not require any ground truth knowledge of task boundaries.

## 5 FULLY ONLINE META-LEARNING WITHOUT TASK BOUNDARIES

We first discuss the intuition behind how our approach handles online meta-learning without task boundaries. In many real-world tasks, we might expect the tasks in the online data stream to change gradually. This makes it very hard to draw a clear boundary between the tasks. Therefore, it is necessary to relax task boundary assumption if we want a robust online learner that can work on a real-world data stream. Additionally, since nearby data points are most likely to belong to the same or similar task, we would expect adaptation to each new data point to be much faster from a model that has already been adapted to other recent data points.

FOML maintains two separate parameter vectors for the online updates ($\phi$) and the meta updates ($\theta$). Both parameterize the same architecture, such that $f_\phi$ and $f_\theta$ represent the same neural network, but with different weights. The online model continuously reads in the latest datapoints from the online data stream, and updates the parameters $\phi$ in online fashion, without any boundaries or resets. However, simply updating the online model on each data point naïvely will not meta-train it to adapt more quickly, and may even result in drift, where the model forgets prior data. Therefore, we also incorporate a regularizer into the online update that is determined by a concurrent meta-learning process (see Fig. 4). Note that FOML only incorporates the meta-parameters into the online updates via the meta-learned regularization term, without ever resetting the online parameters back to the meta-parameters (in contrast, e.g., to FTML (Finn et al., 2019)).

The meta-updates of previous MAML-based online meta-learning approaches involve sampling data from all of the tasks seen so far, and then updating the meta-parameters $\theta$ based on the derivatives

of the MAML objective. This provides a diverse sampling of tasks for the meta update, though it requires storing all of the seen data (Finn et al., 2019). We also use a MAML-style update for the meta parameters, and also require storing the previously seen data. To this end, we will use $\mathcal{B}$ to denote a buffer containing all of the data seen so far. Each new datapoint is added to $\mathcal{B}$ once the label is observed. However, since we do not assume knowledge of task boundaries, we cannot sample entire tasks from $\mathcal{B}$, but instead must sample individual datapoints. We therefore adopt a different strategy, which we describe in Section 5.2: as shown in Fig 4, instead of aiming to sample in complete *tasks* from the data buffer, we simply sample random past datapoints, and meta-train the regularizer so that the online updates retain good performance on *all* past data. We find that this strategy is effective at accelerating acquisition of future tasks in online meta-learning settings where the tasks are not mutual exclusive. We define both types of updates in detail in the next sections.

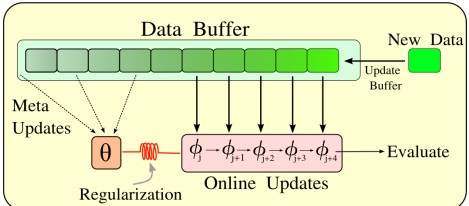

**Figure 2: Overview of FOML learning**: FOML updates the online parameters $\phi$ using only the most recent $K$ datapoints from the buffer $\mathcal{B}$. Meta-learning learns a regularizer, parameterized by meta-parameters $\theta$, via second-order MAML-style updates. The goal of meta-learning is to make $\phi$ perform well on randomly sampled prior datapoints *after performing $K$ steps with the meta-trained regularizer*.

## 5.1 FULLY ONLINE ADAPTATION

At each time step, FOML observes a data point $x_t$, predicts its label $\hat{y}_t$, then receives the true label $y_t$ and updates the online parameters. In practice, we make updates after observing $N$ new datapoints ($N = 10$ in our experiments), so as to reduce the variance of the gradient updates. We create a small dataset $\mathcal{D}_{tr}^j$ with these $N$ datapoints for the time step $j$. The true label for these datapoints can be from class labels, annotations, rewards, or even self-supervision, though we focus on the supervised classification setting in our experiments.

However, the online updates are based only on the most recent samples, and do not make use of any past data. Therefore, we need some mechanism for the (slower) meta-training process to "transfer" the knowledge it is distilling from the prior tasks into this online parameter vector. We can instantiate such a mechanism by introducing a regularizer into the online parameter update that depends on the meta-parameters $\theta$, which we denote as $\mathcal{R}(\phi, \theta)$. While a variety of parameterizations could be used for $\mathcal{R}(\phi, \theta)$, we opt for a simple squared error term of the form $\mathcal{R}(\phi, \theta) = (\phi - \theta)^2$, resulting in the following online update at each step $j$:

$$\phi^j = \phi^{j-1} - \alpha_1 \nabla_{\phi^{j-1}} \{ \mathcal{L}(\phi^{j-1}; \mathcal{D}_{tr}^j) + \beta_1 \mathcal{R}(\phi^{j-1}, \theta) \}$$
$$= \phi^{j-1} - \underbrace{\alpha_1 \nabla_{\phi^{j-1}} \mathcal{L}(\phi^{j-1}; \mathcal{D}_{tr}^j)}_{\text{task specific update}} + \underbrace{2\alpha_1 \beta_1 (\theta - \phi^{j-1})}_{\text{meta directional update}}$$

In the case of classification, $\mathcal{L}$ is the cross-entropy loss. $\alpha_1, \beta_1$ are hyperparameters. Next, we discuss how these meta-parameters are trained so as to maximize the effectiveness of this regularizer at accelerating adaption to new tasks.

## 5.2 META-LEARNING WITHOUT TASK BOUNDARIES

As discussed in the previous section, the online updates to $\phi^j$ include a regularizer $\mathcal{R}$ that transfers knowledge from the meta-parameters $\theta$ into each online update. Additionally, our method maintains a buffer $\mathcal{B}$ containing all data seen so far, which is used for the meta-update. In contrast to prior methods, which explicitly draw a training and validation set from the buffer (i.e., a query and support set) and then perform a separate "inner loop" update on this training set (Finn et al., 2019), our meta-updates recycle the inner loop adaptation that is already performed via the online updates, and therefore we only draw a validation set $\mathcal{D}_{val}^m$ from the buffer $\mathcal{B}$. Specifically, we sample a set of $N$ datapoints at random from $\mathcal{B}$ to form $\mathcal{D}_{val}^m$. We then update the meta-parameters $\theta$ using the gradient of the loss on $\mathcal{D}_{val}^m$ *and* the regularizer $\mathcal{R}$ after the last $K$ updates on $\phi$. In other words, we adjust the meta-parameters in such a way that, if an online update is regularized with this meta-weights, then the loss on the online update will be minimized. This can be expressed via following meta update:

$$\theta = \theta - \alpha_2 \nabla_\theta \left\{ \mathcal{L}(\phi^j; \mathcal{D}_{val}^m) + \beta_2 \sum_{k=0}^{K} \mathcal{R}(\theta, \phi^{j-k}) \right\}$$

Here, $\theta$ and $\phi^j$ are only related via regularization term, unlike FTML (Finn et al., 2019), which sets $\phi^0 = \theta$ at every task boundary.

The choice of sampling $\mathcal{D}^m_{val}$ at random from $\mathcal{B}$ has several interpretations. We can interpret this as regularizing $\phi$ to prevent the online parameters from drifting away from solutions that also work well on the entire data buffer. However, this interpretation is incomplete, since the meta-update doesn't simply keep $\phi$ close to a single parameter vector that works well on $\mathcal{D}^m_{val}$, but rather changes the regularizer *so that gradient updates with that regularizer* maximally improve performance on $\mathcal{D}^m_{val}$. This has the effect of actually accelerating how quickly $\phi$ can adapt to new tasks using this regularizer, so long as past tasks are reasonably representative of prior tasks. We experimentally verify this claim in our experiments. Note, however, that this scheme does assume that the tasks are not mutually exclusive. In future work, it would also be interesting to explore more sophisticated strategies for sampling $\mathcal{D}^m_{val}$, for example by leveraging temporal coherence and drawing $N$ sequential points, which are more likely to belong to the same task. We summarize the complete algorithm in Algorithm 1.

---

**Algorithm 1** Online Meta Learning with FOML

---

1: **procedure** META TRAINING
2: **Require:** $\theta, \mathcal{B}, \phi$, Buffer $\mathcal{B}$, Data stream $\mathcal{S}$
3:     $\phi^0 \leftarrow \phi$
4:     **while** Data stream $\mathcal{S}$ available **do**
5:         $\mathcal{D}^j \leftarrow \mathcal{S}$                        ▷ get new data from online data-stream
6:         $\mathcal{B} \leftarrow \mathcal{B} + \mathcal{D}$                            ▷ add new data to the buffer
7:         $\mathcal{D}^j_{tr}, \mathcal{D}^j_{val} \leftarrow \mathcal{D}$            ▷ partition the data into train and validation splits
8:         $\hat{y}_{tr} \leftarrow f_{\phi^{j-1}}(\mathcal{D}^j_{tr})$                   ▷ make predictions on the train set
9:         $\phi^j \leftarrow \phi^{j-1} - \alpha_1 \nabla_{\phi^{j-1}} \{\mathcal{L}_{task}(\phi^{j-1}; \mathcal{D}^j_{tr}) + \beta_1 \mathcal{R}(\phi^{j-1}, \theta)\}$
10:        $\hat{y}_{val} \leftarrow \phi^j(\mathcal{D}^j_{val})$          ▷ Evaluate the updated model on the validation set
11:        $\mathcal{D}^m_{val} \sim \texttt{random-sample}(\mathcal{B})$          ▷ sample random batch from buffer
12:        $\theta \leftarrow \theta - \alpha_2 \nabla_\theta \{\mathcal{L}_{task}(\phi^j; \mathcal{D}^m_{val}) + \beta_2 \sum_{k=1}^K \mathcal{R}(\phi^{j-k}, \theta)\}$
13:        $j \leftarrow j + 1$

---

## 6 EXPERIMENTAL EVALUATION

Our experiments focus on online meta-learning tasks based on supervised classification. In these settings, an effective algorithm should adapt to changing tasks as quickly as possible, rapidly identifying when the task has changed and adjusting its parameters accordingly. Furthermore, a successful algorithm should make use of past task shifts to meta-learn effectively, thus accelerating the speed with which it adapts to future task changes. In all cases, the algorithm receives one data point at a time, and the task changes periodically. In order to allow for a comparison with prior methods, which generally assume known task boundaries, the data stream is partitioned into discrete tasks, but our algorithm is not aware of which datapoint belongs to which tasks or where the boundaries occur. The prior methods, in contrast, *are* provided with this information, thus giving them an advantage. We first describe the specific task formulations, and then the prior methods that we compare to.

Online meta-learning algorithms should adapt to each task as quickly as possible, and also use data from past tasks to accelerate acquisition of future tasks. Therefore, we report our results as a learning curve, with one axis corresponding to the number of seen tasks, and the other axis corresponding to the cumulative error rate on that task. This error rate is computed using a held-out validation data for each task after the adaptation that task. We evaluate prior online meta-learning methods and baselines on two different datasets (Rainbow-MNIST and CIFAR100). TOE (train on everything), TFS (train from scratch), FTL (follow the leader) and FTML (follow the meta-leader) (Finn et al., 2019) are the baseline methods we compare against our method FOML. See Section 4 for more detailed description of these methods.

**Datasets:** We compare TOE, TFS, FTL, FTML and FOML on two different datasets. Rainbow-MNIST (Finn et al., 2019) was created by changing the background color, scale and rotation of the MNIST dataset. It includes 7 different background colors, 2 scales (full and half) and 4 different rotations. This leads to a total of 56 number of tasks. Each individual task is to classify the images into 10 classes. We use the same partition with 900 samples per each task, as in prior work (Finn

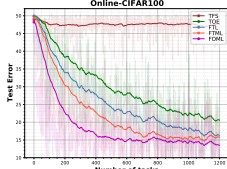 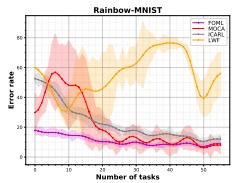 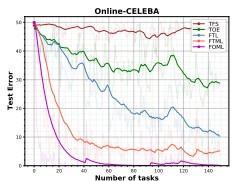

**Figure 3:** *Comparison between online algorithms:* We compare our method with baselines and prior approaches, including TFS (Train from Scratch), TOE (Train on Everything), FTL (Follow the Leader) and FTML (Follow the Meta Leader). **a:** Performance relative to the number of tasks seen over the course of online training on the Rainbow-MNIST dataset. As the number of task increases, FOML achieves lower error rates compared to other methods. **b:** Error rates on the Online-CIFAR100 dataset. Note that FOML not only achieves lower error rates on average, but also reaches the lowest error (of around 17%) more quickly than the other methods. **c:** We also compare our method with continual learning baselines LwF Li & Hoiem (2017), iCaRL Rebuffi et al. (2016) and MOCA Harrison et al. (2020). MOCA Harrison et al. (2020) archive similar performance to ours at the end of the learning, FOML can converge quickly, and faster compared to MOCA. **d:** Performance of FOML on CELEBA dataset. This dataset contains more than 1000 classes and we follow the same protocol as in Online-CIFAR100 experiments. Our method converges faster than other baseline methods on CELEBA with 150 number of tasks.

et al., 2019). However, this task contains relatively simple images, and only 56 tasks. To create a much longer task sequence with significantly more realistic images, we modified the CIFAR-100 dataset to create an online meta-learning benchmark, which we call online-CIFAR100. Every task is a randomly sampled set of classes, and the goal is to classify whether two images in this set belongs to same class or not. Specifically, each task corresponds to 5 classes from CIFAR-100, and every datapoint consists of a *pair* of images, each corresponding to one of the 5 classes for that task. The goal is to predict whether the two images belong to the same class or not. Note that different tasks are not mutually exclusive, which *in principle* should favor a TOE-style method, since meta-learning is known to underperform with non-mutually-exclusive tasks (Yin et al., 2019). To make sure the data distribution changes smoothly over tasks, we only change a subset of the classes between consecutive tasks. This allow us to create a very large number of tasks (1200), and evaluate our method and the baselines on much longer and more realistic task sequences.

**Implementation Details:** We use a simple 4 layer convolutional neural network with 8,16,32,64 filters at each layer for both experiments. However, for the CIFAR-100 experiments, a Siamese version of the same network is used. All the methods were trained via a cross-entropy loss, with their best performing hyper-parameters, on a single NVIDA-2080 GPU machine. Please see Appendix A.1 for more details on hyper-parameters and network architecture.

**Results on Rainbow-MNIST:** As shown in Fig 3, FOML attains the lowest error rate on most tasks in Rainbow-MNIST, except a small segment in the very beginning. The performance of TFS is similar performance across all the tasks, and does not improve over time. This is because it resets its weights every time it encounters a new task, and therefore cannot not gain any advantage from past data. TOE has larger error rates at the start, but as we add more data into the buffer, TOE is able to improve. On the other hand, both FTL and FTML start with similar performance, but FTML achieve much lower error rates at the end of the sequence compared to FTL, consistently with prior work (Finn et al., 2019). The final error rates of FOML are around 10%, and it reaches this performance significantly faster than FTML, after less than 20 tasks. Note that FTML also has access to task boundaries, while FOML does not.

**Results on Online-CIFAR100:** We use a Siamese network for this experiment, where each image is fed into a 7-layer convolutional network, and each branch outputs a 128 dimensional embedding vector. A difference of these vectors are fed into a fully connected layer for the final classification. Each task contains data from 5 classes of CIFAR-100, and each new task introduces three new classes, and retains two of the classes from the previous task, providing a degree of temporal coherence while still requiring each algorithm to handle persistent shift in the task. Fig 3 shows the error rates of various online learning methods, where each method is trained over a sequence of 1200 tasks. All the methods start with initial error rates of 50%. The tasks are not mutually exclusive, so in principle TOE can attain good performance, but it makes the slowest progress among all the methods, suggesting that simple pretraining is not sufficient to *accelerate* learning. FTL uses a similar pre-training strategy as TOE. However it has an adaptation stage where the model is fine-tuned on the new task. This allows it to make slower progress. As expected from prior work (Finn et al., 2019), the meta-learning procedure used by FTML allows it to make faster progress than FTL. However, FOML makes faster progress on average, and achieves the lowest final error rate ($\sim 15\%$) after sequence of 1200 tasks.

## 6.1 ABLATION STUDIES

We perform various ablations by varying the number of online parameters used for the meta-update $K$, importance of meta-model to analysis the properties of our method.

**Number of online parameters used for the meta-update:** Our method periodically updates the online weights and meta weights. The meta-updates involves taking $K$ recent online parameters and updating the meta model via MAML gradient. Therefore, meta-updates depend on the trajectory of the online parameters. In this experiment, we investigate how the performance of FOML changes as we vary the number of parameters used for the meta-update ($K$ in Algorithm 1). Fig **??** shows the performance of our method with various values of $K$: $K = [1, 2, 3, 5, 10]$. We can see that the performance improves when we update the meta parameters over longer trajectory of online parameters (larger $K$). We speculate that this is due to the longer sequences providing a clearer signal for how the meta-parameters influence online updates over many steps.

**Importance of meta update:** FOML keeps track of separate online parameters and meta-parameters, and each of them is updated via corresponding updates. However, only the online parameters $\phi$ are used for evaluation, while the meta-parameters $\theta$ only influence them via the regularizer, and have no direct effect on evaluation performance. This might raise the question: how important is the contribution of the meta-parameters to the performance of the algorithm during online training? We train a model with and without meta-updates, and the performance is shown in Fig **??**. None that, the model without meta-updates is identical to our method, except that the meta-updates themselves are not performed. We can clearly see that the model trained with meta-updates preforms much better than a model trained without meta-updates. The model trained without meta-updates generally does not improve significantly as more tasks are seen, while the model trained with meta-updates improves with each task, and reaches significantly lower final error. This shows that, even though $\theta$ and $\phi$ are decoupled and only connected via a regularization, the meta-learning component of our method really is critical for its good performance.

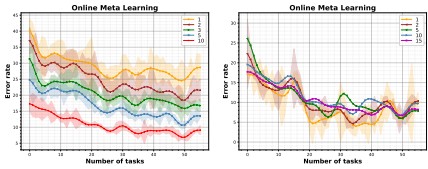

**Figure 4:** *Ablation experiments:* **a)** We vary the number of online updates $K$ used before the meta-update, to see how it affects the performance of our method. The performance of FOML improves as the number of online updates is increased. **b)** This experiment shows how FOML performs with and without meta updates, to confirm that the meta-training is indeed an essential component of our method. With meta-updates, FOML learns more quickly, and performance improves with more tasks.

## 7 CONCLUSION

We presented FOML, a MAML-based algorithm for online meta-learning that does not require ground truth knowledge of task boundaries, and does not require resetting the parameter vector back to the meta-learned parameters for every task. FOML is conceptually simple, maintaining just two parameter vectors over the entire online adaptation process: a vector of online parameters $\phi$, which are updated continually on each new batch of datapoints, and a vector of meta-parameters $\theta$, which are updated correspondingly with meta-updates to accelerate the online adaptation process, and influence the online updates via a regularizer. We find that even a relatively simple task sampling scheme that selects datapoints at random from a buffer of all seen data enables effective meta-training that accelerates the speed with which FOML can adapt to each new task, and we find that FOML reaches a final performance that is comparable to or better than baselines and prior methods, while learning to adapt quickly to new tasks significantly faster. While our work focuses on supervised classification problems, a particularly exciting direction for future work is to extend such online meta-learning methods to other types of online supervision that may be more readily available, including self-supervision and prediction, so that models equipped with online meta-learning can continually improve as they see more of the world.

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

# A  APPENDIX

## A.1  IMPLEMENTATION DETAILS-RAINBOWMNIST

All the images from Rainbow-MNIST are in $28 \times 28$ pixel resolution and with 3 channels. We used a 4 layer convolutional neural network for these experiments, with each layer having [32,32,64,64] number of filters. After every convolution ReLU activation and max pooling is applied to the features. At the last layer we take an average pooling. Finally, 64 dimensional features are passed through fully connected layer with a sigmoid activation. Since the Rainbow-MNIST uses MNIST data to classify, the final layer has 10 neurons, and the objective is to correctly classify the digits of different scales and colors into 10 classes (actual numbers irrespective of the rotation and background color). All the models were trained with 50 gradient updates.

## A.2  IMPLEMENTATION DETAILS-ONLINE-CIFAR100

All the images from CIFAR100 are in $32 \times 32$ pixel resolution and with 3 channels. We used a similar 7 layer convolutional neural network in a siamese network architecture for these experiments, with each layer having [32,32,32,64,64,64,128] number of filters. After every convolution ReLU activation and batchnorm is applied to the features. Every other layers have max-pooling. At the last layer we take an average pooling. The L2 distance between two images are calculated and a fully connected layer is used to classify the two images as same class or not.

## A.3  BASELINE METHODS

**TFS:** Every time this model sees a new task (This needs to know the task boundary), the model resets to new sets of random weights. After that the model is trained only using the data from the new task. At the same time, we also rest the optimizer state to remove any effect of momentum from previous updates. The model is trained with Adam optimizer with a learning rate of 0.001.

**TOE:** Although TOE optimize the parameters on the all available task, training the model using all the available data is practically impossible. Therefore, we sample datapoints from the buffer and update the model parameters. However, since the number of data is increasing over time, the model also needs be updated on a good representative data of the online stream. Therefore, we increase the number of gradient updates over time. For ONLINE-CIFAR100 experiment, the number of gradients updates are increased by 10 for every 100 tasks. The model is trained with Adam optimizer with a learning rate of 0.001.

**FTL:** This method, first pretrains a model on the past data, and then fine-tune it on the very recent samples. Similar to TOE, we first train a set of weights using the datapoints in the data buffer. After that, the model is fine-tuned on the correct task data. However, after this adaptation and evaluation the fine-tuned weights are simply discarded and the pretraining starts from the initial weights of the adaptation process. Adam optimizer with same configuration is used to train this model.

**FTML:** The FTML updates the meta-parameters using MAML style update. For that, we trained the FTML with 5 inner-loop updates (each inner-loop have task-specific data for the inner-loop adaptation). After the meta-update, the meta-parameters are used as the initialization for the inner-loop adaptation, and similar to FTL the adapted weights are simply discarded after evaluation. The inner-loop is trained with SGD with a learning rate of 0.001 and the outer-loop is trained with a learning rate of 0.0005.

**FOML:** Our method also have two sets of parameter vectors, thus two different optimizers. We use Adam for both online and meta updates with a learning rate of 0.001 for both cases. Since, our online updates share the meta-knowledge via the regularization term, the $\beta_1$ controls how much pull is applied to the meta parameters by the online parameters. We use 0.01 for $\beta_1$. Similarly the pull on meta-parameters by the online parameters is controlled by $\beta_2$, and we use 0.001 in our experiments.

