# OpenReview forum: "Fully Online Meta-Learning Without Task Boundaries"
_ICLR.cc/2022/Conference — ICLR 2022 Submitted_

### Official Review · Reviewer_dcpm · 2021-11-01

**Correctness:** 3
**Technical Novelty And Significance:** 2
**Empirical Novelty And Significance:** 2
**Recommendation:** 5
**Confidence:** 3

**Main Review:**

Strengths

1. The idea uses online learning to remove the task boundary is interesting
2. The paper is reasonably written with clear background and diagrams for the overall architecture. It’s well written and easy to follow.
3. The experiments demonstrate that FOML outperformed FTML

Weakness
1. The idea is not new, authors are missing some closely relevant and major literature in online meta-learning [1] and meta-learning without task boundaries.[2]
2. I might have misunderstood the definition of ‘reset’ in the last paragraph of Section 4; as stated FTML needs to be reset at each task and the meta-training process does not influence the current task at all; But in algorithm 1 of the paper FTML, the model uses the result of meta-update to evaluate performance at data points within round t, and meanwhile the result of the adaptation process is recorded. In algorithm 1, why calculate predictions on the train set and validation set? Are they for loss? I do not see the utilization of them. And in lines 9 and 12, what’s the meaning of \L_{task}? Is this loss related to specific tasks?  The FTML uses \phi to meta-update \theta and then uses the \theta to update \phi. The proposed method utilizes the \theta to regularize and update \phi, and then update \theta by \phi. It is not clear to me how the task boundaries are removed. More explanations are expected.
3. The experiments are weak. For example, there is no performance evaluation of the proposed FOML in Figure 3 which makes your claim unsupportive. I assume the OML represents the FOML? I’m not sure about that.
4. According to my previous point, important baselines are missing, the work [2] also targets meta-learning without the task boundaries.
5. Is there any mathematical guarantee to show that FOML can learn without task boundaries? If not, more experiments on extra datasets are expected to support this claim.
6. Some minor presentation issues, e.g.,
- In Implementation Details, there is an error of “Please see Appendix ??”
- For figure 3, the method name should be FOML rather than OML in the diagram.
- In A.2 Implementation Details-Online CIFAR100, all the images should from ‘CIFAR100’ instead of Rainbow-MNIST.
- For the caption of figure 1, online parameters should be \phi and meta-parameters should be \theta.


Reference

[1] Denevi, G., Stamos, D., Ciliberto, C. and Pontil, M., 2019. Online-within-online meta-learning. In ADVANCES IN NEURAL INFORMATION PROCESSING SYSTEMS 32 (NIPS 2019) (Vol. 32, pp. 1-11). Neural Information Processing Systems (NeurIPS 2019).
[2] Harrison, J., Sharma, A., Finn, C. and Pavone, M., 2020. Continuous Meta-Learning without Tasks. Advances in Neural Information Processing Systems, 33.


**Summary Of The Paper:**

This paper proposes an online meta learning method which can remove the discrete task boundaries. Different from existing online meta-learning method, FOML utilizes online learning to remove the task boundaries.

The FOML is able to continually update online parameters with new datapoint, and meanwhile perform meta-gradient updates on a set of meta-parameters using a buffer of previous data. The authors compare the model to state-of-the-art like FTML on benchmarks such as Rainbow-MNIST and CIFAR100 datasets. Experimental results show that FOML learns to adapt faster and can achieve lower error rates on classification tasks. However, it lacks sufficient experiments and comparison with baseline methods to support the claim that FOML can learn without the task boundaries.


**Summary Of The Review:**

This paper proposes a method named FOML which is a further step of the existing method FTML that tries to remove the task boundary. However, such an idea is not new and there is already a similar work that exists. Experiments are not strong enough to support the claim.

---

> ### Author Response · Authors · 2021-11-23
> **We thank the reviewer for the valuable feedback.**
>
> We thank the reviewer for the valuable feedback. We have revised the paper with the comments from the reviewer, including comparison of our method on recent meta learning baseline MOCA (Harrison et al. 2019), and two other incremental learning baselines, we also added a new large scale dataset CELEB A for comparison, and revised the paper with additional literature section and limitations.
>
> * Related works - Thank you for these references, they are definitely relevant. We added a comparison to MOCA [2], which we discuss below under “Experiments.” We have also added a discussion about [1], [2] in the related work section. Specifically, Online-within-Online Meta learning [1] indeed solves tasks in an online manner, however it requires the knowledge of the task boundaries to update their meta model. In contrast, our work focuses on learning online meta updates, without any additional knowledge about the task boundaries, with a method that is substantially different from [1]. MOCA [2] does avoid the need for ground truth knowledge of task boundaries, but it instead attempts to estimate the task boundaries explicitly. While this is feasible when tasks change suddenly (i.e., when there is a change point), it may be difficult when the changes are more gradual. For example, in our OnlineCIFAR dataset, the task is changing very slowly by adding one or two classes at a time. In contrast, our method neither requires knowledge of the task boundaries nor attempts to estimate where they are, but instead adopts a fully online strategy that dispenses with the need for boundaries entirely. We believe that this is significantly novel, but we would appreciate pointers to any other relevant work that we missed.
>
> * Resetting -  \L_{task} is simply the loss we are optimizing for the model, for example for the RainbowMNIST experiment it is cross entropy loss, and for the OnlineCIFAR dataset it is binary cross entropy loss.  \L_{task} is not specific to every task in the dataset. However, it might depend on the dataset and the goal of the dataset.
>
> * why calculate train/val? - the performance evaluation is just for evaluating the method, and the graphs are taken from those evaluated values. It is not used in the algorithm for any updates.
>
> * FTML - It uses \theta to learn a meta model and then adapt from \theta towards \phi. The initial value of \phi is initialized as \theta and fine tuned during the adaptation. On the other hand, in FOML \phi and \theta are totally two different models (not an initialization of the other), and they only communicate via the regularization term. This way FOML does not reset and initialize oine model with another model’s parameters.
>
> * task boundaries - After every adaptation, FTML discards the learned \phi parameters and starts adopting from the meta parameters \theta again. This is what we define as resetting. For FTML to start adopting and resetting, it should know the task boundaries. However, FOML does not adopt \phi from \theta, therefore there is no resetting involved. The meta parameters control the trajectory of \phi via regularization only.  This allows us to overcome the requirement for the task boundary knowledge.
>
> * Experiments - We have added additional experiments on a large scale CELEB dataset (See Fig 3 in the revised paper). It has more than 1000 classes and we evaluated our method and other baselines on this dataset. Our method performs better than other baselines and converges faster. We also added an experiment on RainbowMNIST with two continual learning baselines (iCaRL, LwF). As the reviewer suggested, we also include the comparison with the MOCA [2] paper on RainbowMISNT tasks (See Fig 3 in the revised paper). While MOCA and FOML achieve similar performance at the end of the learning process, our method learns faster compared to MOCA.
>
> * Theoretical guarantees - While we agree that better theoretical guarantees would be valuable, to our knowledge virtually all prior work on meta-learning (much less online meta-learning) does not provide theoretical guarantees that the method will work. Indeed, even for supervised deep learning providing such guarantees is a major open problem. Therefore, while this is of course important, we believe it is reasonable to leave this major open problem outside of the scope of the current paper.

---

> > ### Comment · Reviewer_dcpm · 2021-11-30
> > **RE: Author's Response**
> >
> > Thanks for adding those related works that I raised in the comments.
> > I do find the experimental results for MOCA in Rainbow-MNIST, as MOCA address a very similar problem with yours, I would like to see more comprehensive experiments for MOCA in other two datasets - Online-CIFAR100 and Online-CELEBA.
> > I'm willing to raise my score to 5 as some of my concerns are remaining.
> >
> > Typos:
> > Oonline-within-Online --> Online-within-Online

---

> > > ### Author Response · Authors · 2021-11-30
> > > **Thanks for your response**
> > >
> > > Dear Reviewer,
> > >
> > > We will definitely add comparisons to MOCA on the other tasks in the final -- there is nothing preventing us from doing this, but unfortunately during the rebuttal period, we simply did not have the time to run these experiments to completion, since the CVPR deadline was in the middle of the rebuttal period, and these experiments take quite a bit of time to complete. Is this the only remaining issue with the paper? If so, this is an easy thing to add, it's just a matter of waiting for the experiments to complete.

---

### Official Review · Reviewer_me6U · 2021-11-02

**Correctness:** 2
**Technical Novelty And Significance:** 2
**Empirical Novelty And Significance:** 2
**Recommendation:** 6
**Confidence:** 4

**Main Review:**

Strengths

The approach is quite simple and intuitive.

The paper is written clearly and is easy to understand.

Concerns

Lack of clarity on the problem statement: The authors cast the problem as meta-learning that must be done online. I am not convinced with this problem statement. As I understand, the goal in meta-learning (particularly the MAML framework adopted by the authors) is to learn a good initialization for quickly adapting to a new task. However, the current problem seems to be more on the lines of online learning and avoiding catastrophic forgetting, as there is no adaptation step after meta-training. The authors explicitly mention that their goal is not to solve catastrophic forgetting. But then, what is the goal? If I am mistaken, and there is an adaptation step(akin to meta-test), it would be interesting to see the number of adaptation steps required by the various models to adapt to the new task.

Missing literature: Online learning/continual learning is a well-studied research area. There is a large body of literature on regularization-based methods (e.g., Kirkpatrick et al. 2017, Ritter et al., 2018), memory-based methods, online continual learning (Caccia et al. 2020), etc. I find the discussion on the related work to be quite narrow, missing out on approaches trying to achieve similar goals. Furthermore, evaluating these related works over and above the few methods listed in the experiments is also important.

Weak experimental setup: The specially curated datasets (and the tasks) appear quite simple (even the online-CIFAR100 task). Existing literature has used more complex sequences of tasks (from miniimagenet, tieredimagenet). Experiments on these datasets are warranted to conclude FOML’s efficacy.

Finally, the lack of theoretical justification on the effectiveness of the FOML is also a cause of concern.

Minor comments

Page 5 last paragraph - To this end, we well \script{B} to denote - we use \script{B} to denote.

Implementation Details Section on Page 8 - Appendix information is missing.


**Summary Of The Paper:**

The paper proposes an online learning approach that utilizes the meta-learning framework. The proposed method - FOML has two main components: a regularizer that does not allow the updated parameters of the model to deviate too far from the previously learned set and a meta update on the parameters using the online updates, and a validate set drawn from the buffer of examples seen in the past. FOML is evaluated on two datasets and compared against a few other similar approaches and baselines. The results suggest faster adaptation of FOML compared to previous methods.


**Summary Of The Review:**

Overall, while the paper is well-written, I have doubts about the motivation behind the problem statement. I am concerned about the missing relevant literature and inadequate experiments.

--- Post rebuttal update

I thank the authors for the detailed response to clarify some of my concerns. I also read the other reviews and the response from the authors. The authors have clarified the existence of the problem statement citing prior work. I agree with the existence of the problem statement. However, I am not entirely convinced with the motivation. Experiments on real-world applications requiring online meta-learning would have significantly strengthened the contribution. I acknowledge that the experiments comparing other online-learning algorithms (such as LwF, iCaRL) and algorithms such as MOCA is a step in the right direction, but requires more analysis (using more datasets).

---

> ### Author Response · Authors · 2021-11-23
> **We thank the reviewer for the valuable feedback.**
>
> To address the reviewer's concerns about the experiments, we added an additional set of experiments on the large-scale CELEB dataset, comparison with MOCA (Harrison et al. 2019) and to two prior continual learning methods (iCaRL and LwF). These additional results are in the revised manuscript. We've also included the suggested citations in the related work section, and we clarify some of the misunderstanding about the method's goals below. Please let us know if this addresses the concerns raised in your review.
>
> * Experiments - We have added additional experiments (See Fig 3 in the revised paper) on a large scale CELEB dataset, which has more than 1000 classes. Our method performs better than other baselines and converges faster. We also added comparisons to two prior continual learning methods (iCaRL and LwF) on the RainbowMNIST dataset. We hope that these additional experiments address your concerns about the completeness of experimental results, though we would be happy to receive any more suggestions about suitable tasks and comparisons.
>
> * Problem statement - The goal of our method is to reduce the number of mistakes made while learning a sequence of tasks from streaming data. This problem statement is similar to the one proposed by Finn et al. (“Online Meta-Learning”). This problem statement exists in prior work, it is not new, although our approach additionally relaxes the assumption Finn et al. makes about knowledge of ground truth task boundaries. There is indeed both an adaptation step and a meta-learning step, but these steps are done in parallel (continually) on streaming data. The main innovations of our approach compared to past work on online meta-learning (e.g., Finn et al. FTML) is to avoid the need to reset the adapted parameters between tasks, and also to completely remove any notion of explicit task boundaries. Empirically, our method attains better results than prior methods, even when those prior methods have access to ground truth task boundaries.
>
> * Literature - We thank the reviewer for pointing out these works, we have included these papers in our revision. However, we also emphasize that our work does not focus on solving catastrophic forgetting (similarly to past work on online meta-learning, see e.g. Finn et al. “Online Meta-Learning”). There are major differences between our line of work and continual learning literature. Our goal is to learn new tasks as fast as possible, while the continual learning is focused on learning new tasks without forgetting previously learned tasks. We evaluate our method only on the new task performance while the continual learning methods measure the performance across all the seen tasks.
>
> * Theoretical guarantees - While we agree that better theoretical guarantees would be valuable, but to our knowledge virtually all prior work on meta-learning (much less online meta-learning) does not provide theoretical guarantees that the method will work well. Indeed, even for supervised deep learning providing such guarantees is a major open problem. Therefore, while this is of course important, we believe it is reasonable to leave this major open problem outside of the scope of the current paper.

---

### Official Review · Reviewer_XzKp · 2021-11-03

**Correctness:** 3
**Technical Novelty And Significance:** 2
**Empirical Novelty And Significance:** 2
**Recommendation:** 6
**Confidence:** 3

**Main Review:**

Strength:
1. Overall speaking, the writing is clear for people to understand the motivation, the background, the proposed method, and experiments.
2. The algorithmic changes to previous FTML are incremental (to suit the new assumption) but the justification behinds these design choices are reasonable.
3. Experiments on synthetic datasets validated the design choices to some degree, as the proposed approach achieves better fitting performances.

Weakness/Question to Authors (order does not matter):
- I am less familiar with the literature and would like to understand the evaluation metric on both datasets. How was the error rate / test error being calculated? Are they essentially the regret that represents your fitting performance? Do you also measure backward transfer performances and forward transfer performances?
- In meta-update, with a large K, you essentially need to store K copies of all network parameters? Then how is the storage cost comparing your method to previous approaches?
- Why don't you also compare to other non-meta-learning based online continual learning? This would help people to understand the status quo of meta-learning based online continual learning. It is okay even though you can not outperform them.
- What would be the optimal fitting performance? Say, for all  the1200 tasks, you first perform offline batch training on all of them sufficiently, and then fine-tune on each of those 1200 tasks? How close would your model's performance to this optimal performance (of the same model achitecture)?
- Did you compare different strategy for the buffer, such as reservoir buffer vs. fIFO buffer vs other buffer?
- Why don't you consider realistic online continual learning datasets, such as [1] and [2]?

  - [1] Drinking from a Firehose: Continual Learning with Web-scale Natural Language.
  - [2] Online Continual Learning with Natural Distribution Shifts: An Empirical Study with Visual Data

Minor Comments:
1. Section 1 Paragraph 1: "… simultaneously that data for each new task is also used" missing verb after simultanously?
2. Section 1 Paragraph 1: "fall short of the goal of creating an effective and adaptation system" -> "effective adaptation system"?
3. Section 3 Paragraph of "Online Learning": I am not sure why loss function is also a sequence of time step. Then, what would the loss function be if there is an concrete example. Isn't the loss function determined by the data (and label)? Maybe I misunderstand?
4. Equation of FTL, maybe it would be more clear to reader if you rewrite phi as phi_{TOE}^{t-1}.
5. Figure 3. Does not your approach called FOML? The method name in the legend is OML
6. Implementation Details. Broken appendix hyperlink.

**Summary Of The Paper:**

This paper extends meta-learning algorithm for continual online learning. Different from previous meta-learning based online continual learning, this algorithm removes assumption of knowing task boundaries of data beforehand. The proposed approach extends the previous online MAML (FTML), by removing the discrete 'resets' of task-specific parameters. One key difference besides removing reset is to change the replay buffer (memory) strategy used by meta-updates, where prior approach always sample data of all previous tasks and the proposed approach samples all data points seen so far (so there is no need to know which task we are currently solving).

As a result, this adapted algorithm cope with data of continuous task boundaries. To validate the proposed approach, authors conducted experiments on a synthetic continual image recognition benchmark (of 1,200 tasks) based on CIFAR100, where they outperforms previous continual meta-learning based approach.

**Summary Of The Review:**

I am not an expert in this area so I believe that my opinion should be discounted. Based on my understanding, I think this is a solid and well-written paper, with extensive results that demonstrates the improvement of presented approach to its comparing prior work.

---

> ### Author Response · Authors · 2021-11-23
> **We thank the reviewer for the valuable feedback.**
>
> We thank the reviewer for the valuable feedback. We have revised the paper to address the various comments and suggestions (added new large scale dataset for comparison, more baselines (MOCA Harrison et al. 2019) and revised the literature review), and we detail specific answers to individual questions below.
>
> * Other datasets - We used the usual benchmarking datasets for online learning (Finn et al), and since our goal is not focused on continual learning, we did not test our method on continual learning datasets. However we have added additional experiments with a large scale dataset (CELEBA). The experimental results show that in the CELEB dataset also FOML performs better than the baseline methods.
>
> * Other baselines -  In our experiments, most of our baselines except FTML are actually non-meta-learning approaches. Additionally, these methods are compared  in the recent online meta learning works (Finn et al). We included an experiment with MOCA (Harrison et al. 2019) as reviewers suggested. Our results show that, even though both FOML and MOCA achieve similar performance at the end, FOML was able to converge quickly (See Fig 3). We also add two more incremental learning approaches in our experimental setting. (iCaLR, iTAML) and FOML performs better than the continual learning baselines.
>
> * Evaluation - In the Rainbow-MNSIT experiment, the task is to classify the correct digit with different background colors and rotations. Therefore, in every task there will be all the 10 digits in the dataset. The accuracy is calculated as the 10-way classification accuracy. For the CIFAR100 and CELEBA experiments, the task is to classify whether two images belong to the same class or not. This metric is the same as the metric used in prior work (FTML, Finn et al.), and therefore we believe it to be the most reasonable choice for evaluating an online meta-learning method, though we would welcome suggestions about other metrics to report. In regard to backward transfer: although we do not evaluate backward transfer explicitly, our task distribution is stationary, so it is reasonable to expect that the time needed to adapt to a previously seen task will not be larger than for a new task. Note, however, that the aim of online meta-learning (see, e.g., FTML by Finn et al.) is not to memorize the set of previously seen tasks, but rather to adapt as quickly as possible to new tasks, so if the goal is primarily to enable backward transfer, there are likely other approaches that would be better.
>
> * Memory - Our method does not need to store K copies of the network. However, it does need to store K copies of the activations in order to calculate the meta-gradient. We agree with the reviewer that memory is of course an issue, but this is an issue with all the gradient-based meta learning algorithms, including the standard MAML algorithm, and is not unique to our method. In our experiments we use K=5, which is in the same order on inner loop updates as MAML and FTML.  We have addressed this limitation in the paper.
>
> * Optimal fitting - Thanks for this comment, our FTL is somewhat similar to the oracle performance baseline. At the end of the learning curve, FTL has been pre trained on all the seen tasks and fine tuned on the current task. This is equivalent to the optimal fitting performance the reviewer is suggesting. At the end, all the methods will converge to this point as long as they see enough data. However, our method can reach this point faster.
>
> * Buffer - Our buffer is just to keep the data in the memory. It does not affect the performance of any method.

---

### Official Review · Reviewer_dEVK · 2021-11-19

**Correctness:** 2
**Technical Novelty And Significance:** 3
**Empirical Novelty And Significance:** 2
**Recommendation:** 6
**Confidence:** 5

**Main Review:**

# Strengths
* The proposed method is simple and it performs better than FTML and other previous approaches.
* In general, the text is clear and easy to read.
* The authors provide an Algorithm.
* Implementation details are provided in the Appendix.

# Weaknesses
* Since FOML continuously updates the online weights, it is not clear what would happen if an OOD task is encountered. It would be interesting to see some ablation experiment or some discussion about this scenario, as well as possible ways to deal with it.
* Some examples of non-cited works that tackle this problem are [A] or [B], which try to address this problem and also introduce new versions of MAML. In fact, it seems like [A] could be highly related to your approach. In their case, rather than optimizing a regularizer, they optimize the inner loop learning rate, which in the end, has the effect of preventing the online weights from drifting too much when the new encountered task has a high interference.

[A] Gupta, Gunshi, Karmesh Yadav, and Liam Paull. "La-maml: Look-ahead meta learning for continual learning." arXiv preprint arXiv:2007.13904 (2020).
[B] Caccia, Massimo, et al. "Online fast adaptation and knowledge accumulation (osaka): a new approach to continual learning." Advances in Neural Information Processing Systems 33 (2020).


## Typos
* Pag 4. algorithms tries -> algorithms try
* Pag 4. with a pretrained weights -> with pretrained weights
* Pag 5. necessarily results -> necessarily result
* Pag 5. we will B to denote -> we will use B to denote
* Pag 7. changings -> changing
* Pag 8. see Appendix ??
* Pag 9. depends on -> depend on

**Summary Of The Paper:**

The authors propose FOML, a new online learning algorithm based on MAML. It introduces two new features. 1. Instead of resetting the inner loop weights back to the meta-weights after each task, the online weights are always kept and updated. 2. It introduces a regularizer that aims to keep the online weights always close to the meta-weights and the other way around. Rainbow MNIST and online CIFAR100  experiments show that FOML outperforms TFS, TOE, FTL, and FTML. In ablation experiments, the authors show that FOML woks better for higher numbers of inner update iterations as well as the importance of the meta-update.

**Summary Of The Review:**

The authors present a simple approach that is sound, effective in a specific online continual learning scenario, where a replay buffer is kept, and new tasks are not completely unrelated to previous tasks. On the other hand, the authors only provide experiments on MNIST and CIFAR, and do not provide any additional information on how their algorithm would handle more challenging situations.

Overall, the proposed method is interesting for the research community but the authors should provide more information on how it would behave under more challenging scenarios, discuss its limitations, and include in the text a more complete comparison with other setups that justifies for a non-expert reader why they did not compare with Harrison et al. 2019, He et al. 2019, [A] ...

---

> ### Author Response · Authors · 2021-11-23
> **We thank the reviewer for the valuable feedback.**
>
> We thank the reviewer for the valuable feedback. We have revised the paper with the comments from the reviewer. To address your concerns, we've added discussion of the two suggested prior papers, and also added three new baselines and one new dataset to the experiments to address your concerns about the difficulty and completeness of the experimental results. We discuss the experiments in depth below, followed by a discussion of the related works (which we now cite) and a brief comment about OOD tasks.
>
> * Experiments - We have added additional experiments on a large scale CELEB dataset (See Fig 3 in the revised paper). It has more than 1000 classes and we evaluated our method and other baselines on this dataset. Our method performs better than other baselines and converges faster. We also added an experiment on RainbowMNIST with two continual learning baselines (iCaRL, LwF). As the reviewer suggested, we also include the comparison with the MOCA(Harrison et al. 2019) paper on RainbowMISNT tasks (See Fig 3 in the revised paper). While MOCA and FOML achieve similar performance at the end of the learning process, our method learns faster compared to MOCA.
>
> * Related works - We thank the reviewers for pointing out these works, we have added these works in the related works and discuss them. We also added a comparison with Harrison et al. 2019 and show that our method indeed learns faster. We also included a discussion in the related works section about [A] and [B] works. While [A, B] both address the continual learning from the lens of meta learning, as mentioned in the paper our goal is not to minimize catastrophic forgetting in this paper. We are interested in quickly learning new tasks using the knowledge learned in the past tasks. We also added two continual learning baselines (iCaRL, LwF) in the paper and compared them against our work. The experimental results show that, FOML works better than the continual learning baselines. See Fig 3 in the revised paper.
>
> * OOD tasks - Handling OOD tasks is an important topic in meta-learning, but handling entirely OOD tasks is outside the scope of this paper. We've added a brief comment about this scope in Section 2, and added a sentence about OOD tasks in Section 7, indicating that this is an important direction for future work. That said, our experiments do already contain some amount of distributional shift: in each setting, the tasks change gradually over time, introducing modest amounts of distributional shift, and since our method uses gradient-based adaptation, we would expect that in the case of an unusual task, it would still make forward progress, albeit slowly. However, we do not want to make specific claims about this, since it is outside of the scope of our paper, and instead leave an in-depth investigation of meta-learning under distributional shift to future work.

---

> > ### Comment · Reviewer_dEVK · 2021-11-29
> > **Response to Authors**
> >
> > Dear authors,
> >
> > Thanks for your comments, which address most of my concerns. I have raised my score to "weak accept" in accordance.
> >
> > However, there are two issues I would like to highlight:
> >
> > 1. It is not clear why you chose CelebA for the new experiments.
> > 2. The text in the revised version of the pdf is of worse quality than the original. Some words are misspelled (gradinet), and citations are inline. I suggest you put those in parentheses or use `natbib` and `\citep` if possible. Could you also add (a), (b), (c) .. below the different subfigures in Figure 3?

---

### Decision · Program_Chairs · 2022-01-20

**Decision:**

Reject

**Comment:**

The paper propose a Fully Online Meta-Learning (FOML) method which extend MAML for continual learning in a fully online learning  without requiring the knowledge of the task boundaries. Experiments show that FOML was able to learn new tasks faster than several existing online learning methods on Rainbow-MNIST, and CIFAR100 datasets.

There are a few major concerns from reviewers. One concern is about the lack of clarity on the problem statement: The authors cast the problem as meta-learning that must be done in a fully online setting, but it requires to store all the training data in a buffer storing all the training data seen so far, which contradicts to the principle of “online learning”. Another major weakness is the poorly written literature survey, which missed to cite a large body of related work in continual learning and online-meta-learning (such as Online Continual Learning, task-free continual learning, continual learning without task boundaries, etc). These should at least be discussed carefully if not fully compared in the empirical studies. Also experiments are quite weak in both settings, datasets and rather out-of-date baselines. Finally, there also lacks of theoretical justification or analysis.

Therefore, the paper is not recommended for acceptance in its current form. I hope authors found the review comments informative and can improve their paper by addressing these review comments carefully in future submissions.